# Intertwined and Finely Balanced: Endoplasmic Reticulum Morphology, Dynamics, Function, and Diseases

**DOI:** 10.3390/cells10092341

**Published:** 2021-09-07

**Authors:** Hannah T. Perkins, Viki Allan

**Affiliations:** 1Division of Molecular and Cellular Function, School of Biological Sciences, Michael Smith Building, The University of Manchester, Dover Street, Manchester M13 9PT, UK; hannah.perkins@manchester.ac.uk; 2Biological Physics, Department of Physics and Astronomy, Schuster Building, The University of Manchester, Oxford Road, Manchester M13 9PL, UK

**Keywords:** endoplasmic reticulum (ER), morphology, dynamics, anomalous diffusion, membrane contact site (MCS), dynein, kinesin, microtubule

## Abstract

The endoplasmic reticulum (ER) is an organelle that is responsible for many essential subcellular processes. Interconnected narrow tubules at the periphery and thicker sheet-like regions in the perinuclear region are linked to the nuclear envelope. It is becoming apparent that the complex morphology and dynamics of the ER are linked to its function. Mutations in the proteins involved in regulating ER structure and movement are implicated in many diseases including neurodegenerative diseases such as Alzheimer’s, Parkinson’s, and amyotrophic lateral sclerosis (ALS). The ER is also hijacked by pathogens to promote their replication. Bacteria such as *Legionella pneumophila* and *Chlamydia trachomatis,* as well as the Zika virus, bind to ER morphology and dynamics-regulating proteins to exploit the functions of the ER to their advantage. This review covers our understanding of ER morphology, including the functional subdomains and membrane contact sites that the organelle forms. We also focus on ER dynamics and the current efforts to quantify ER motion and discuss the diseases related to ER morphology and dynamics.

## 1. Introduction

The endoplasmic reticulum (ER) is a subcellular organelle responsible for a variety of essential cellular functions. In the ER, proteins are synthesised and undergo post-translational modification, calcium ions are stored and released as dictated by cellular signaling, and lipids are biosynthesised. The ER is the largest organelle in the cell and extends as a continuous membrane-bound entity from the nuclear envelope to the cell periphery in a complex network of tubules and sheet-like regions. Fast dynamics and re-organisation, along with the fact that the ER comprises ~35% of the cytoplasmic volume, mean that the ER explores the volume of the cytoplasm more rapidly than any other organelle [1]. Little is understood as of yet about the function of ER dynamics or the effects of dynamics in ER-related diseases.

The complex structure of the ER is maintained by a variety of proteins including reticulons, REEPs, and atlastins [2,3,4]. These morphology-regulating proteins promote membrane curvature [2,3], drive tubule fusion [4,5,6,7,8], stabilise junctions [9,10,11], regulate sheet spacing [12,13,14], and tether the ER to microtubules [15,16,17]. Cooperation between these proteins is essential for normal morphology, with overexpression of certain types leading to abnormal proportions of sheets and tubules in the network [6,9,18]. The microtubule motor proteins kinesin-1 and cytoplasmic dynein (referred to herein as dynein) also bind to the endoplasmic reticulum to drive the formation of new tubules and to continuously and dynamically remodel the network (e.g., [19,20]). Interactions between the ER and other organelles at membrane contact sites (MCSs) also contribute to the dynamics of the organelle (Section 3.1.2). ER dynamics is an emerging topic of research. Historically, tubule extension speeds have been measured [20,21,22,23], but the quantification of other aspects of ER dynamics has been challenging due to limitations in microscopy and computation. Recently, software designed to analyse the motion of the ER in live cells has been published [24,25,26,27] and further developments in this area are likely in the near future.

Although the relationship between ER dynamics and function is not yet understood, abnormal ER dynamics have recently been linked to several diseases [8,28,29], indicating that dynamics are likely to be very important. In contrast, disease-related mutations in morphology-regulating proteins are rather better understood and links between morphology and function have been established [30,31].

This review will detail the current understanding of ER morphology, dynamics, and function and the interplay between them, as well as the diseases linked to abnormalities in morphology and dynamics, focussing primarily on the ER in vertebrate cells. Section 2 will describe ER morphology, the proteins involved in maintaining normal ER structure, and the links between morphology and function that have been elucidated. Section 3 will describe the dynamics of the ER, factors causing organelle motion, current efforts to quantify the dynamics, and recent developments in computational analysis and modelling. Finally, Section 4 will detail the known links between ER structure/motion and diseases as well as the proteins involved in these diseases. A discussion of the subdomain organisation of the ER specifically in neuronal cells is presented in another article of this special issue [32].

## 2. Morphology

A lipid bilayer bounds the lumen of the ER. This bilayer is formed into sheets and tubules of roughly 80–100 nm diameter [25,33,34], with a range of ~50–100 nm [33], which are connected at junctions to form a reticular network. Tubules are around 1 µm in length on average and connect at junctions, usually joining 3 tubules [27,35]. The sheets are flat, with a bilayer separation of 30–50 nm [33], and often occur in a stacked configuration with helical ramps connecting the layers [36]. Sheet density is usually higher nearer the centre of the cell (e.g., [33,37,38]), where the endoplasmic reticulum also includes the nuclear envelope; however, sheet-like regions also occur in peripheral ER. Super-resolution light microscopy has revealed that some peripheral structures originally thought to be sheets actually consist of dense matrices of tubules [33,34,38], while others comprise sheets fenestrated with tiny holes (nanoholes: [33]). Research into the proteins involved in shaping the ER membrane is extensive and has been reviewed recently [30,31,39]. The complex morphology of the ER has recently been linked to its function (reviewed in [40]) and can be changed in response to altered metabolic requirements or cell cycle stage, as described later.

### 2.1. Morphology-Regulating Proteins

A plethora of proteins are involved in regulating ER morphology. These proteins stabilise high membrane curvature regions, promote membrane fusion, maintain three-way junctions, regulate sheet thickness and flatness, and anchor the ER to microtubules (Figure 1).

ER tubules and sheet edges are regions of high membrane curvature. The hairpin-like REEP and reticulon protein families promote curvature in the surface of the ER to stabilise tubules and sheet edges [2,3]. Reticulon 4 also accumulates at the edges of nanoholes, which are absent in Rtn4 knockout cells [33]. These proteins are thought to sit with the wider end of the ‘hairpin’ at the outer edge of the membrane to promote curvature via hydrophobic wedging, as shown in Figure 1C.

Membrane fusion is also essential to maintain correct ER morphology. Atlastin, an ER-resident GTPase, is required for ER network maintenance and membrane fusion [4,5,6]. New tubules that are drawn out from the existing membranous structure must be fused to the network to preserve normal ER morphology. Bian et al. proposed that atlastin is resident in the membrane of both of the uniting edges and that a dimer is formed between the two proteins [41] (see Figure 1F). This dimerisation would cause a conformational change in which the proteins, and therefore the ER membrane, would be pulled closer together, eventually causing membrane fusion. Two RabGTPases, Rab10 [7], and Rab18 [8] have also been implicated in ER tubule fusion. Depletion of either of these proteins caused an increase in sheets and a decrease in tubules. Rab10 also localises to the growing tips of ER tubules that are enriched in two lipid synthesis enzymes: phosphoinositol synthase (PIS) and CEPT1 [7]. Interestingly, dynamin-related protein 1 (Drp1), a protein required for mitochondrial fission, has been shown to lead to promote ER tubule formation independently of Rtn4 [42].

The three-way junctions of the ER network in mammalian cells are stabilised by mammalian Lnp1, a member of the Lunapark family [9,10,11]. mLnp1 localises to three-way junctions in Cos-7 cells, and junctions lacking in mLnp1 were much more dynamic and shorter-lived than their mLnp1-enriched counterparts [10]. Lunapark-free junctions frequently disappeared by ring closure, in which junctions slide along a neighbouring tubule until they coalesce with another junction, whereas ring closure was less common for mLnp1 junctions [10]. Therefore, Lunapark stabilises three-way junctions in the network.

The cooperation of these proteins is crucial in ER morphology regulation. In order to maintain normal ER morphology, a balance between reticulon 4a and atlastin expression was required [6]. Long, unbranched tubules were seen in cells with little atlastin function and in cells overexpressing reticulon 4a. Both atlastin and reticulon 4a expression in Lunapark knockout cells restored normal network morphology at the cell periphery. In yeast, a balance between the Lunapark Lnp1p and the atlastin homologue Sey1p is required for typical polygonal network formation [9]. Lunapark and atlastin also coordinate to form and maintain three-way junctions in mammalian cells, with Lunapark unable to localise to junctions when atlastins are deleted [18]. Using a different but complementary approach, Shemesh et al. modelled the ER network, focusing on two classes of morphology regulating proteins [43], with Lunapark being proposed to be a negative-curvature-inducing protein and reticulons enhancing straight edges. Altering the expression level of Lunapark in Cos-7 cells gave rise to different ER morphologies, and various network properties such as the total length and the preferred shape of junctions were replicated by changing protein concentrations in the simulations. Clearly, the relative concentrations of the ER morphology-regulating proteins are of great importance for normal ER structure.

CLIMP-63, an integral membrane protein, has been suggested as a mechanism of controlling the lumenal separation between the two membranes of ER sheets [12,13,14]. As shown in Figure 1A, its large coiled coil domain sits within the lumen of the ER and has been hypothesised to bind to the coiled coil domains of other CLIMP-63 proteins to regulate sheet thickness [12]. Over-expression of CLIMP-63 induces the formation of sheets containing nanoholes [33]. The second factor influencing sheet thickness may be groups of ribosomes, also called polyribosomes, bound to the cytosolic face of the ER [44] (see Figure 1G). Removing ribosomes from the ER surface converted sheets to tubules, supporting the hypothesis that polyribosomes stabilise ER sheets [45]. p180 and kinectin, two proteins that are similar in conformation to CLIMP-63, may play a role in flattening the surface of sheets [13], although there is little conclusive evidence supporting this idea. Both proteins are potential receptors for the microtubule motor kinesin-1 (see Section 3.1.1). Microtubules and the actin cytoskeleton play a major role in regulating ER position and morphology, both via dynamic and static links (Figure 1D,E), as discussed in Section 3. Together, the proteins involved in cytoskeletal binding and motility, membrane curvature stabilisation, tubule fusion, junction maintenance, and sheet thickness regulation are essential for the maintenance of normal ER structure.

### 2.2. Structural and Functional ER Subdomains

Different structural domains of the ER, which are responsible for specific functions, have been hypothesised since the first electron microscopy images of the ER were obtained. In the 1950s, it was noticed by George E. Palade that ER sheets tended to be studded with ribosomes, known as rough ER, whereas tubules were largely ribosome-free, or smooth [46,47]. This difference in structure in different regions of the network sparked the hypothesis of ER functional subdomains. Since then, significant work has been undertaken to correlate the morphology of the ER with its function.

#### 2.2.1. Protein Factory and Quality Control

The best-understood role of the ER is to synthesise and insert proteins into the ER membrane or lumen, and this happens primarily by ribosomes associating with the cytosolic face of the ER, where the newly synthesised polypeptide chain is translocated through the Sec61 translocon, as reviewed in this special issue by Sicking et al. [48]. In *S. Cerevisiae*, sheet ribosome density was found to be significantly larger than tubule ribosome density [49]. As ribosomes attach to the ER to allow the translation of secreted or membrane-bound proteins, this result, along with the results of electron microscopy studies such as those by Palade [46,47], led to the idea that sheets are the main site for protein biosynthesis in the ER. The relatively low membrane curvature and large lumenal volume of sheets are optimal for both the binding of ribosomes or polyribosomes to the bilayer and for the accessibility of chaperones to nascent peptides needed for folding and post-translational modifications. In agreement with the idea that ribosome-studded sheets of rough ER form a “protein factory” functional domain, cells that are specialised in secreting proteins, such as pancreatic secretory cells, have a higher proportion of ER sheets than cells that secrete very few proteins, such as epithelial cells and neurons [50]. Components of the translocon were also enriched in sheets [13]. Taken together, these results strongly suggest that sheets are the primary location of protein biosynthesis in the ER.

Newly synthesised membrane and lumenal proteins need to fold properly and be appropriately modified post-translationally, for example by the addition and subsequent trimming of glycan chains and the formation of disulphide bridges (reviewed in [51,52]). This process involves multiple ER-resident chaperones and enzymes which act in sequence to assist and monitor the correct folding and glycosylation status through the calnexin/calreticulin cycle. Any proteins that fail to fold are recognised and removed from this cycle, to be retrotranslocated out of the ER and degraded by the proteosome in a process termed ER-associated degradation (ERAD) [51]. The load of misfolded proteins is closely monitored, and if too many build up, the unfolded protein response (UPR) is triggered, which leads to upregulation of the key ER-resident proteins required for protein folding combined with inhibition in protein synthesis [53]. While the proteins involved in folding and glycan addition are ubiquitously distributed throughout the ER, there is evidence that certain key proteins in the quality control and ERAD pathways may be concentrated in specialised structures called the ER-derived quality control compartment (ERQC) which is localised next to the nucleus [52]. While the exact nature of this compartment and how it is connected to the bulk ER remains unclear, its localisation depends on the microtubule motor dynein [54].

Interestingly, prolonged ER stress has been seen to induce the reversible formation of whorls of ribosome-free ER membranes that contained the Sec61 translocon and the key UPR signalling enzyme PKR-like ER kinase (PERK), but not reticulons, CLIMP63 or the lumenal marker calreticulin [55]. The whorls formed from vesicular/tubular structures that budded from the ER via the COPII pathway (see below) and subsequently fused and flattened. This is markedly different to normal conditions, where Sec61 is excluded from COPII vesicles. These whorls may facilitate two UPR outcomes: inhibition of protein translocation by segregating and inactivating translocons, and activation of PERK. The whorls resemble the organised smooth ER (OSER) previously seen when certain ER proteins such as HMG-CoA reductase or cytochrome b5 are over-expressed [56,57]. How whorls and OSER relate to the ERQC is a key question to be addressed in the future.

#### 2.2.2. ERES: Export Checks

Correctly folded lumenal and membrane proteins leave the ER at ER exit sites (ERES), which are structurally distinct, ribosome-free puncta located in the rough ER network. Exit sites consist of a cluster of vesicular-tubular membranes [58,59], continuous with the ER membrane. In vertebrate cells, ERES are scattered throughout the network and protein transfer from the ER to the Golgi relies on microtubule-dependent transport via the dynein/dynactin motor protein complex [60]. The ERES themselves undergo short-range movements on microtubules [61]. Two protein coat complexes, COPI and COPII, aid in the formation and organisation of the exit sites as well as in protein transport. COPII forms a scaffold to deform the membrane, regulates cargo entry into ERES [62], and remains localised to ERES even after cargoes have departed [63,64,65]. COPI however, travels with the cargo as it is transported away from exit sites [66], as does Rab1 [65]. The exact roles of COPI in cargo trafficking away from the ER are unknown, but it may play a role in sorting and delivering cargo to the Golgi apparatus [66,67].

The higher-order structure of ERES has only recently been identified using FIB-SEM. Weigel et al. discovered that an interwoven network of narrow tubules (40–60 nm in diameter) exist at exit sites, connected to the ER by a slightly narrower COPII neck [63]. Long, pearling tubules with COPI punctae were also found to extend from the exit sites, along microtubules towards the Golgi apparatus. Pearled outlines are a hallmark of longitudinal tension in tubular membranes [68] and may be the result of forces applied by dynein/dynactin [60]. Such membrane pearling has been hypothesised as a precursor to fission, transforming tubules into vesicles [69]. Dynein may be recruited to ERES membranes via an interaction between dynactin p150 and the COPII components Sec23 and Sec24 [70], but how the motor attaches to carriers once the coat is lost is an open question. One potential route is via BicD2 and Rab6, which have been shown to recruit dynein to ERES and drive their concentration in the perinuclear region [59]. Kinesin-1 is also present on both ERES [61] and the transport carriers they produce [71]. Interestingly, exit sites were also found to double in diameter in response to the accumulation of cargo, while the diameter of nearby ER tubules was unaffected [63]. Together, these studies show that ER exit sites are highly organised cargo export subdomains, capable of responding to changes in cell requirements.

#### 2.2.3. MCSs: Lipid Manufacture

Given that sheets are likely to be responsible for protein biosynthesis, could tubules be responsible for lipid synthesis and calcium ion homeostasis? In support of this idea, cells responsible for steroid synthesis, such as adrenal cortical cells, have proportionally more ribosome-free smooth ER [72]. It has also been suggested that membrane contact sites (MCSs) between the ER and other organelles may be more common in the ribosome-free tubular region of the network, particularly those MCSs formed with the plasma membrane, endosomes, lipid droplets and mitochondria [73,74,75,76,77,78]. The MCSs and relevant proteins discussed in this review are shown in Figure 2. MCSs are tethered connections between organelles, where the membranes are not fused, but separated by 5–30 nm [79,80,81,82]. Such close proximity of the membranes enables the non-vesicular transfer of cargo between the organelles [83]. MCSs are present between the ER and almost every other subcellular organelle [38,81]. Their function, morphology, and dynamics are diverse and well-reviewed (e.g., [84,85,86]). Here, we will focus on the links between MCSs and ER function and dynamics.

Lipid biosynthesis is known to occur primarily in the ER [87,88]. Using cell fractionation, enrichment of the machinery involved in lipid and sterol synthesis was discovered in the ER membrane that is associated with mitochondria [89,90,91] and the plasma membrane [92]. More recently, it has been shown that ER–plasma membrane contact sites are required for the synthesis of phosphatidylcholine, an integral component of biological membranes, in yeast [93]. Lipid metabolising machinery has also been found at the leading edge of dynamic ER tubules [7] and in dynamic ER-derived vesicles that contact many other organelles in Cos-7 cells [94]. The ER is also responsible for the synthesis of neutral lipids [95]. Neutral lipids lack charged groups and therefore cannot easily be integrated into lipid bilayers. Instead, lipid droplets are formed, which consist of a phospholipid monolayer surrounding a core of neutral lipids. Exact details about the formation of lipid droplets are still up for debate; however, it is now known that lipid droplets are formed in the ER [96,97,98]. Neutral lipids gather within the lipid bilayer of the ER, forming a lens between the inner and outer leaflets [97], before budding off into the cytoplasm [99]. In plants, it has also been suggested that lipid droplets are formed in specialised subdomains of the ER [100]. In addition to being formed in the ER, lipid droplets can re-contact the ER once they have budded off [101]. The ER–lipid droplet membrane bridges formed are proposed to facilitate the exchange of lipid metabolising enzymes, without which large lipid droplets are not formed [102,103,104]. The discovery of upregulated lipid metabolising machinery in the tubular ER, particularly at MCSs with the plasma membrane, mitochondria and lipid droplets, suggests that these subdomains also participate in lipid metabolism [7,89,90,91,92,94,95].

#### 2.2.4. MCSs: Lipid Exchange

Newly-synthesised lipids must be delivered from the lipid synthesis subdomains of the ER to their final destination. Each membrane-bound organelle has a specific lipid composition [107] and therefore lipid transfer requirements are tailored to each organelle. Lipid composition affects many properties of an organelle membrane, including its curvature and the proteins recruited to both the cytoplasmic face of the organelle and the transmembrane proteins [108]. Lipids are known to be transported from the ER to other organelles via vesicular and non-vesicular transport mechanisms, such as lipid transfer proteins.

Lipid transfer from the ER to the Golgi apparatus occurs via a vesicular mechanism, which forms part of the secretory pathway, and via monomolecular lipid transport mediated by lipid transfer proteins at the ER–Golgi interface. In the vesicular transport route, the membrane leaves the ER via ERES (Section 2.2.2) and the same vesicles/tubular-vesicular clusters that carry ER-synthesised proteins from the ER to the Golgi can also contain lipids metabolised in the ER [109]. These lipids then continue along the secretory pathway to reach their destination.

Lipid transfer proteins traffic lipids between organelles that are not connected by vesicular transport pathways [83]. These transfer proteins are enriched at membrane contact sites between the ER and the plasma membrane [110,111,112,113,114,115,116], Golgi apparatus [117,118,119], and endosomes [120,121,122,123,124,125] (lipid transfer proteins at ER MCSs were recently reviewed in [108]). Evidence has recently been discovered for three-way MCSs between the ER, late endosomes, and mitochondria, via PDZD8, a protein that possesses a lipid transfer domain and interacts with protrudin, a key component for ER motility (Section 3.1.2), and Rab7 [126]. This three-way contact is hypothesised to facilitate lipid transfer between these three organelles. Contacts between the ER and late endosomes/lysosomes involve the ER-anchored VAP proteins (e.g., [82,127,128,129]), and these interactions are sensitive to nutrient status [128,129]. Importantly, contact sites between early endosomes and the ER in low cholesterol conditions facilitate the transfer of cholesterol from the ER to the multivesicular body, where it is needed to drive endosomal sorting via the formation of intralumenal vesicles (ILVs) [120]. As an aside, ER-early endosome contacts also facilitate ILV formation by providing sites where the ER-localised protein tyrosine phosphatase 1B dephosphorylates endocytosed, active growth factor receptors such as the epidermal growth factor receptor, which is required for EGFR to be sorted into ILVs [130]. This may be why motile early endosomes have been seen to pause at ER tubules [74]. Bidirectional cholesterol transfer also occurs at contacts between the ER and late endosomes/lysosomes (reviewed in [86,131]), and these sites involve a number of proteins implicated in recruiting microtubule motors (see below).

Peroxisomes and the ER must exchange lipids as the synthesis of some lipids, for example, ether phospholipids, begins in peroxisomes but is completed in the ER [132,133]. The lipid transfer protein VPS13D has been discovered at both ER–peroxisome and ER–mitochondria contacts, where it interacts with Miro [134], and another, VPS13A, has been found at ER–mitochondria contacts [124], and this lipid transport has been shown to be important for peroxisome biogenesis [135]. The machinery involved in creating ER–peroxisome MCSs has recently been discovered [136] and there is some evidence for non-vesicular ER to peroxisome lipid transport [137]. Similarly, phosphatidylserine must be transferred from the ER, where it is synthesised, to the mitochondria, where it is converted to phosphatidylethanolamine [138,139,140]. It has been shown that this lipid transfer occurs even without cytosolic phospholipid exchange proteins or small vesicles [139,140] and is therefore likely to happen via lipid transfer proteins at ER–mitochondria MCSs. ERMES (ER–mitochondria encounter structure), a protein complex found in yeast, has been proposed as a tether-forming complex between the two organelles [141,142,143,144]. This complex may also transfer lipids at contact sites, via transport proteins such as Lam6/Ltc1. Lam6 interacts with the mitochondrial proteins Tom70 and Tom71 at ER–mitochondrial MCSs and is known to transfer sterols in vitro [145,146,147]. Likewise, PDZD8 may fulfil a similar role [126].

These non-vesicular pathways are a significant mechanism of lipid trafficking. Indeed, it was found that the rate of lipid transfer from the ER to the plasma membrane does not appreciably decrease when vesicular pathways are blocked [148,149,150,151], indicating that non-vesicular transport alone can sustain the required lipid transfer to the plasma membrane. As MCSs between the ER and other organelles, particularly the plasma membrane, mitochondria, and endosomes [73,74,75,76,77,78], preferentially form in the tubular ER network, these findings suggest that lipid transfer occurs primarily in the tubular ER.

#### 2.2.5. MCSs: Calcium Control

Another important function of the ER is calcium ion sequestration and release. Ca^2+^ is an important signalling molecule, the concentration of which affects not only the function of the ER but also a wide range of other pathways, including mitochondrial metabolism and apoptosis [152,153,154]. Chaperone proteins within the ER such as calnexin [155], calreticulin [156], and protein disulphide isomerase [157], among others, bind to Ca^2+^ and their function as chaperones in protein folding is dependent on the calcium ion concentration in the ER [158,159]. The accumulation of improperly folded proteins leads to ER stress and activates the unfolded protein response (UPR), which can either restore ER homeostasis or induce apoptosis, depending on the cellular circumstances (reviewed in [160]). Therefore, regulation of Ca^2+^ concentration is imperative for normal cell function.

Calcium ions are released from the ER by transmembrane receptors, primarily the ryanodine receptor (RyR) and the inositol 1,4,5-trisphosphate receptor (IP_3_R), in response to intracellular cues. Mitochondria are located in positions close to IP_3_Rs [161,162] in order to take up Ca^2+^ ions upon their release. Calcium ions are necessary for mitochondrial metabolism, including ATP production and reduction of pyridine nucleotides [152,153]. In order to transfer Ca^2+^, a complex is formed between the ER-resident IP_3_R and voltage-dependent anion channel 1 (VDAC1) in the outer mitochondrial membrane [161,163,164]. Grp75, a cytosolic protein, forms a tether between the channels to facilitate Ca^2+^ transfer to the mitochondria [165]. Mitofusin 2 is also implicated in both ER–mitochondrial tethering and mitochondrial calcium uptake [105], although there is some debate as to its exact role (reviewed in [166]). MCSs between the ER and mitochondria, which primarily occur in the tubular ER, are clearly of great importance for calcium ion homeostasis [85]. IP_3_Rs also mediate calcium transfer to lysosomes at ER–lysosome MCSs [167].

Upon ER calcium ion depletion, an influx of Ca^2+^ from extracellular sources is required to replenish the lumenal Ca^2+^ concentration. Transport of Ca^2+^ into the cell is accomplished by the cooperation of STIM1 and Orai1. After calcium stores have been depleted, the ER-resident protein STIM1 [168] and the plasma membrane calcium channel, Orai1 [169], are recruited to ER-PM MCSs [170] where they form a complex [171]. Orai1 is a calcium release-activated calcium (CRAC) channel which is opened when the interaction with STIM1 occurs [172]. This process of Ca^2+^ influx is known as store-operated calcium entry (SOCE). The Ca^2+^ entering the cell is then transported into the ER via sarco/endoplasmic-reticulum Ca^2+^ ATPase (SERCA) pumps. These pumps expend ATP to transport calcium ions against the Ca^2+^ concentration gradient into the ER, refilling lumenal calcium stores [173,174,175]. From the work summarised here, it is apparent that MCSs with mitochondria and the plasma membrane are responsible for calcium ion release from the ER and influx from the extracellular region respectively. STIM1 also plays an important role in ER dynamics, by linking ER tubules to growing microtubules to form tip attachment complexes (TACs), as described in Section 3.1.3.

#### 2.2.6. MCSs: Control of Membrane Fission and Fusion

A fascinating aspect of MCS function is that in several cases, they act as hotspots for membrane fission or fusion of the organelle bound to the ER. For example, mitochondrial fission occurs at points where they contact ER tubules and become constricted before the fission protein Drp1 is enriched [38,73]. Drp1 is always accumulated at, or next to, ER–mitochondrial contacts. Interestingly, Drp1 has recently been shown to facilitate ER tubule formation, and to be localised on all ER tubules at low levels, as well as generating sites for mitochondrial–ER interaction and mitochondrial fission [42]. However, this role in generating ER tubules does not require Drp1′s GTPase activity, whereas that is essential for mitochondrial fission [42]. Mitochondrial fusion has also been shown to occur more frequently when mitochondria are attached to the ER [38].

As mentioned above, the ER also forms contacts with early endosomes, and these can also be the site of endosome fission [176]. One isoform of the microtubule-severing protein spastin localises on the ER membrane, and it interacts with the early endosomal ESCRT protein IST1 at ER-early endosome contacts to drive endosomal tubule fission and sorting. Disrupting this interaction led to the missorting of lysosomal enzymes and lysosomal defects, which is likely to be the underlying reason why spastin mutations cause hereditary spastic paraplegia [177]. Another ER protein, reticulon 3L, has recently been shown to be recruited to ER–endosome contact sites by Rab9 and promote endosome maturation and sorting [178], likely explaining why endosome maturation correlates with enhanced interactions with the ER [75]. It will be interesting to determine if these are the same or different pathways.

More than 90% of late endosomes/lysosomes are associated with the ER [38,179] and 80% of endosomal fission events happen when associated with the ER [176]. Retromer drives the sorting and recycling of material from the late endosome to the Golgi apparatus, and the scission of retromer tubules happens at points of contact with the ER and requires an ER membrane protein, TMCC1 that accumulates at ER–endosome contact sites, the actin-binding protein coronin 1 [179], and the WASH complex [127] and its interactor, strumpelin [177].

## 3. ER Dynamics

The ER is not only complex in its organisation, but also in its motion. Progress in the understanding of ER dynamics has been slower than that of ER morphology as the narrow tubules and constant motion and rearrangement of the network make the ER challenging to image. ER dynamics in mammalian cells can be categorised into three types: oscillation of established network elements; the dynamics of particles within the ER lumen or membrane; and generation of new network elements (see Figure 3). The purpose of ER dynamics is still unclear, but the predominant theory is that oscillations accelerate the processes carried out in the ER by facilitating the movement of lumenal and transmembrane particles [25,180,181].

### 3.1. Cytoskeletal Control of ER Dynamics

The ER constantly rearranges its spatial organisation. The impressive dynamics of the ER were observed in living cultured CV1 cells [182], newt lung cells [23], and growth cones in cultured neurons [183] long before the discovery of green fluorescent protein (GFP), by the use of the lipophilic dye DiOC_6_. New tubules can be drawn out from the existing network and fused to neighbouring tubules or junctions to create new connections, and network polygons can form and disappear ([182]; Figure 4). This microtubule-motor-driven movement is described in Section 3.1.1. The proportion of the network in sheets and tubules can also change dynamically. ER sheets reorganise into tubules when ribosomes are stripped from their surface with puromycin [45]. As described below, inhibiting microtubule-based tubule movement can in turn increase the proportion of sheet-like regions. Entry into mitosis has been reported to trigger sheet expansion [37,43,184], although other studies have demonstrated enhanced mitotic tubulation [45] that is driven by REEPs 3 and 4 [185]. Such dynamic reorganisation may help the organelle to sample the cellular volume rapidly [1] and respond to changes in cellular requirements and nutritional status (Section 3.1.3). ER-associated microtubule motors are not the only means by which the ER interacts with the cytoskeleton, and in the following sections, we describe how tubules can also be extended by interactions with growing microtubules via tip attachment complexes (TACs) and by association with motile MCSs. Static interactions of ER tubules with microtubules and the role played by the actin cytoskeleton are also outlined.

#### 3.1.1. Microtubule Motors Drive ER Dynamics

Early studies using DiOC_6_ revealed that ER tubules in the cell periphery often co-aligned closely with microtubules [23,183,186], as confirmed in many subsequent studies (e.g., [21,38,187]). Ring rearrangement and tubule branching also occur in association with microtubules [38]. The depolymerisation of microtubules completely inhibited ER tubule and network dynamics in VERO cells [20] and increased the amount of ER in sheet-like structures and reduced the tubular network [20,37,38,186,188]. Super-resolution imaging in live cells has shown that these sheets actually consist of a mixture of morphologies, including thick sheets, thinner sheets containing nanoholes, and dense tubular networks [33], suggesting that microtubules not only control the position of the ER network but also contribute to the detailed organisation of ER membrane domains.

Direct visualisation of ER-like tubules extending along microtubules came first from in vitro assays using extracts from CV1 cells [189] or interphase *Xenopus* eggs [190] where video-enhanced differential interference contrast microscopy revealed membrane tubules sliding along microtubules. The *Xenopus* networks were shown to be ER by antibody labelling and the presence of polysomes on the membrane surface [19]. Motility brought tubules in contact with each other, resulting in tubule fusion [19,20,189] that was atlastin-dependent [184]. In addition, both smooth and rough ER from rat liver formed motile networks when combined with interphase *Xenopus* egg cytosol, demonstrating cross-species conservation of motility and membrane fusion [191]. However, if the concentration of membrane is high enough in vitro, an ER tubule network can form in the absence of microtubules [192]. The motor-driven sliding of ER tubules along microtubules has since been visualised many times using fluorescence microscopy (e.g., [20,21,22,38]).

Since microtubules are oriented in most cultured non-neuronal cells with their dynamic ‘plus’ ends towards the cell periphery, rapid outward ER tubule sliding requires a plus-end-directed microtubule motor. The founding member of the kinesin superfamily, kinesin-1, is responsible for this motility [20,193,194]. Inhibition of kinesin-1 not only inhibited outward tubule movement but also increased the proportion of ER sheet regions in the cell periphery [20]. Given that sheet-like regions can consist of either accumulated fine tubular networks [34] or sheets containing nanoholes [33], it will be interesting to see if these sheets correspond with either type of structure, or if there is a mixture of morphologies, as seen after nocodazole treatment [33]. Most kinesin-1 in animal cells is tetrameric, comprising two identical motor (KIF5) subunits and two identical KLCs [195]. Vertebrates express three KIF5 genes: KIF5B is expressed ubiquitously, while KIF5A and C are neuronally enriched. We will return to KIF5A later since mutations cause neurodegenerative diseases (Table 1) [195]. There are four KLC genes, with KLCs 1, 2, and 4 being widely expressed. KLC1 exists in multiple alternately spliced forms, with KLC1B being required for ER motility [20,193].

An important outstanding question is the identity of the kinesin-1 receptor on the ER. Several candidates have been proposed, but there are caveats with all of them. It is possible that there are different receptors in neuronal vs. non-neuronal cells, and in different tissues. Kinectin was identified using a function-blocking monoclonal antibody [196] and binds to the KIF5 C-terminus [197]. Kinectin has been implicated in the regulation of focal adhesion dynamics during chemotaxis by promoting ER targeting to focal adhesions at the cell’s leading edge [198,199]. Interestingly, Rab18 is required for this process and forms a ternary complex with kinectin and kinesin-1 [198]. However, kinectin knock-out mice had no phenotype, with normal ER (and other organelle) distribution in KO MEFs [200]. Furthermore, in cultured neurons, the ER extends all along axons, yet kinectin is only observed in the cell body [194], in keeping with its proposed role in maintaining ER sheet spacing [13]. Nevertheless, the siRNA depletion of kinectin decreased ER network dynamics in Cos-7 cells [26]. Intriguingly, Rab10, is also implicated in regulating ER tubule formation [7] and has been reported to form a complex with JIP1, a neuronally expressed kinesin adaptor, and KLC1, to promote the transport of secretory vesicles during neuronal polarisation and axon growth [201]. Whether this complex is involved in ER tubule extension in neurons, and whether the Rab18-kinectin-KIF5B complex is needed for ER tubule extension in non-migrating cells are open questions.

Another candidate ER kinesin-1 receptor is p180, which shares homology with kinectin’s kinesin-1 binding domain [202], and which, like kinectin, has also been localised to central sheet regions in non-neuronal cells [13,16,194]. In cultured neurons, however, p180 localised not only to sheets in the cell body, but also to tubules in axons, but not dendrites [194], a localisation consistent with KIF5 binding. However, it is likely to be acting as an anchor between the ER and microtubules rather than as a kinesin-1 receptor (see Section 3.1.3).

There are two further candidate kinesin receptors. Protrudin (gene name ZFYVE27) is an ER-resident kinesin binding protein with a key role to play in generating motile late endosome–ER MCS motility, as described in Section 3.1.2. However, it is expressed at very low levels (undetected in the HeLa cell proteome [203]), and while its depletion with siRNA leads to late endosome clustering at the cell centre [204,205], it is not clear what effect this has on ER distribution. Finally, the ER-localised transmembrane DNA-J-domain protein B14 has been shown to interact with KIF5B to generate a site for SV40 virus release from the ER [206], but its involvement in normal ER dynamics remains to be tested.

Given that the ER extends outwards from the nuclear envelope towards the cell periphery, an unexpected finding was that ER tubules moved towards microtubule minus ends in interphase *Xenopus* egg extracts, driven by dynein [19,184,190,207,208]. This fits with the requirement for dynein at the nuclear envelope to drive pronuclear migration, which can be reconstituted in these extracts [209]. However, exclusively dynein-driven ER motility continued even in extracts made from embryos after the fifth cell division: kinesin-dependent ER movement was only seen when cytosol from a tadpole cell line was used [187]. Recent work has provided a satisfying explanation for this phenomenon [210]: the perinuclear pool of ER that accumulates due to dynein activity is needed to assemble the large nuclei seen in early embryos (sea urchin and *Xenopus* embryos in this study). Furthermore, expressing additional reticulon 4b decreased the size of nuclei, presumably by reducing the formation of ER sheet regions [210]. *Xenopus* egg extracts have also revealed cell cycle-dependent changes in ER dynamics, with dynein-driven movement being inhibited in metaphase-arrested extracts [184,190,207] while myosin V-driven ER motility on actin filaments was activated [211]. ER sheets accumulated [184], as has been reported in mitotic HeLa cells [37], although other studies contradict this [45,185].

Dynein is not just an important ER motor in embryonic cells. Around half of rapid ER tubule movements in VERO cells occurred towards the cell centre, and were dynein driven [20]. Furthermore, the inhibition of dynein led to a profound accumulation of ER sheets in the cell periphery without affecting outwards, kinesin-driven movement [20]. Similarly, both dynein and kinesin-1 drive ER tubule motility in axons [194] and dendrites [212,213] of cultured rodent hippocampal neurons. As yet, the receptor for dynein on the ER has not been identified in any system, unlike for ERES (Section 2.2.2).

A final example of microtubule-motor-driven movement involving the ER is provided by nuclear migration and positioning. As well as the pro-nuclear migration mentioned above, kinesin and dynein coordinate nuclear positioning in many different situations, such as during neuronal nuclear migration during brain development [214] and nuclear movement at many stages of *C. elegans* development [215]. Dynein at the nuclear envelope is important for centrosome separation in late G2/prophase, and facilitates nuclear envelope fragmentation (reviewed in [216]). Kinesin-1 is also involved in centrosome and nuclear positioning in non-polarised cells, where it is recruited to the nuclear envelope by RanBP2 and BICD2 [217]. Interestingly, nesprin 4 is specifically expressed at the outer nuclear envelope in polarised epithelia, where it recruits kinesin-1 which then translocates the nucleus to the base of the cell [218].

#### 3.1.2. MCS-Mediated ER Dynamics

In addition to tubule extension driven by motors directly at the ER membrane, ER dynamics can be caused by ER membrane contact sites with early and late endosomes, lysosomes, and mitochondria [22,26,38,74,75,84,85,86,204,219,220] that move along microtubules. This is an example of a process referred to as ‘hitchhiking’, where one organelle provides the motor and drives the movement of another cargo that is not motile by itself [38,221,222]. In fact, early work on ER tubule movement identified morphologically-distinct motile domains at the tips of some moving ER tubules [182,191]. However, there is clear evidence from transmitted light microscopy and DiOC_6_ labelling that ER tubules themselves can translocate directly along microtubules, without the need for hitchhiking with another organelle [19,182,183,190].

Around half of Rab5-positive endosomes in Cos-7 cells are attached to the ER during imaging [75], and when imaged at low frame rates (1 frame per 1.5 s) appeared less motile than ER-associated lysosomes [219], and less likely to cause ER tubule mobility [26]. However, live imaging at the rapid frame rates needed to capture fast early endosome movement has revealed that moving endosomes can translocate towards an ER tubule, grab it and continue moving, pulling out an ER tubule behind them ([74]; Figure 4). As early endosomes move primarily towards the cell centre, driven by dynein [74,223], hitchhiking on early endosomes may account for some of the dynein-dependent ER tubule extensions [20]. How dynein and kinesins are recruited to early endosomes, and the role of kinesin-1 in their movement, is not fully understood.

Recent work suggests that 30–50% of ER tubule extension events are driven by hitchhiking on late endosomes/lysosomes moving along microtubules, while 40% of tubules moved along microtubules independently, and the remainder were mediated by TACs or dTACs (see below) [26,38,219]. Surprisingly, almost all lysosomes were associated with, and moved with, the ER network [219]. Lysosomes that were imaged before, during, and after a hitchhiking event were seen to slow down when attached to ER tubules [204,219], possibly because there was less drag counteracting the force generated by the lysosomal motors when not extending an ER tubule. Two ER-resident proteins involved in phospholipid synthesis, PIS and CEPT1, which have previously been shown to be at the tips of motile tubules along with Rab10 [7], were associated with LE/lysosome hitchhiking events [26].

Again, dynein and kinesin-1 are the major motors driving late endosome/lysosome movement, and there are multiple ways in which they are recruited and controlled, some of which involve interactions with the ER (reviewed in [128,129]). Dynein can be recruited via RILP and Rab7 (in the presence of high cholesterol levels); ALG2 and TRPML1 (regulated by PI(3,5)P_2_ and calcium); or JIP4 and TMEM55B (promoted by starvation, which increases TMEM55B transcription via mTORC1) [128,129]. Kinesin-1 is recruited by SKIP and Arl8 (regulated by BORC), or FYCO1 and Rab7 (regulated by PI(3)P and involving protrudin at the ER: see below), and both linkages are via KLCs [128,129]. An open question is whether these multiple mechanisms function in parallel on the same organelle, or whether there is spatial selection (e.g., on specific membrane subdomains), or switching depending on metabolic status, or other inputs.

If only 30–50% of ER tubules move in association with late endosomes/lysosomes, how important is this ER hitchhiking for ER organisation overall? Two recent studies have demonstrated that in Cos-7 cells at least, the answer is—very. Disrupting ER-late endosome MCSs by RNAi-mediated depletion of VAPA (which binds ORP1L to anchor LE to ER via Rab7: [128]) also disrupted ER morphology and reduced the extent of ER tubules and network complexity, especially at the periphery [26,219], with the knockdown of all ER VAP family members (VAPA, VAPB and MOSPD2) giving a stronger phenotype [26]. In addition, manipulating the motors present on late endosomes/lysosomes provided a useful means of triggering inward or outward endosome movement, which led to a reduction or increase in peripheral ER tubules [26,219] and network dynamics [26]. Interestingly, the depletion of SKIP or Arl8 caused major changes in the ER network, suggesting that this is the major route for kinesin-1 recruitment for this process, rather than protrudin/FYCO1 [219].

Protrudin, the product of the ZFYVE27 gene, is a multispanning transmembrane ER protein that has a plethora of interactors, both at the ER, and at late endosomes. At the ER, it binds to the ER shaping proteins atlastin, REEPs 1 and 5, reticulons 1, 3, and 4, and also interacts with VAP via an FFAT motif ([224]; reviewed in [225]). It can also bind to late endosomes via its Rab-binding domain, which binds Rab7-GTP, and an FYVE (Fab-1, YGL023, Vps27 and EEA1) domain, which binds to the late endosomally enriched phosphotidylinositol 3-phosphate (PI3P) [204,225]. Crucially, it also binds to all KIF5 family members, although the interaction is strongest with KIF5A [226], which is striking considering both protrudin and KIF5A (but not KIF5B or C) can cause hereditary spastic paraplegia when mutated (Table 1). Over-expression of either protein caused the formation of protrusions in non-polarised cells [226], hence protrudin’s name [225], while siRNA-mediated depletion led to an expansion of CLIMP63-labelled sheet-like regions into the cell periphery [224].

Given protrudin’s potential to bind to the late endosome, Raiborg and coworkers investigated if it was involved in MCS formation and found that was indeed the case [204]. Furthermore, overexpression of protrudin led to an accumulation of late endosomes/lysosomes at the cell periphery, a phenotype that had previously been seen for FYCO1, another PI3P and Rab7-binding protein, and which was found to interact with protrudin [204]. Imaging of FYCO1 and protrudin in living cells revealed that moving FYCO1-positive late endosomes interacted with protrudin at the ER, pausing or slowing down while they did this, then detached and moved off more rapidly. Protrudin binds KIF5 and FYCO1 binds KLC, and the expression of protrudin increased the amount of KIF5 found associated with FYCO1, leading to the model that kinesin is passed from protrudin to FYCO1 on the late endosome during ER-late endosome association, so activating late endosome movement once they break free of the ER. While this model is appealing, more formal proof is needed. What is clear though, is that both protrudin and FYCO1 are important for axon extension [129,195,204,227]. Protrudin and FYCO1-mediated late endosome translocation to the cell periphery has also been shown to be important for invadopodia formation, where late endosomes deliver the matrix protease MT1-MMP for secretion, which is necessary for cancer cell migration [205].

Importantly, the late endosome/lysosome position is controlled by nutritional statuses, such as cholesterol and amino acid level, which in turn regulate dynein and kinesin-1 recruitment or activity [128,129,228]. This regulation has recently been shown to have a major impact on ER dynamics and distribution within the cell [26,219]. Serum starvation led to a less mobile ER network and reduced late endosome/lysosome motility, leading to a less complex ER network in the cell periphery with fewer tubule junctions [26]. Lu and coworkers found that a 4 h serum starvation led to late endosome/lysosome clustering, and a reduction in the proportion of tubular ER, as did cholesterol enrichment [219]. In contrast, 24 h starvation or cholesterol depletion triggered peripheral localisation of endosomes with no effect on ER tubules [219].

The protrudin-mediated ER–endosome/lysosome contact pathway is also influenced by nutritional status. The neuronal isoform of carnitine palmitoyltransferase 1, CPT1C, is an ER protein that is regulated by malonyl-CoA levels and mutated in HSP [228]. Recent work has revealed that CPT1C is needed for proper neuronal growth and controls the transport of late endosomes/lysosomes to the axon tip, and this needs its ability to bind malonyl-CoA [228]. It interacts with protrudin, and expressing it in HeLa cells increased the proportion of outward-moving FYCO1-labelled late endosomes if malonyl-CoA was present, but reduced movement to below control levels if malonyl-CoA was depleted. However, unlike protrudin, CPT1C was present, but not enriched, at ER–lysosome contacts, suggesting that it regulates the protrudin–FYCO1–kinesin-1 interaction rather than being directly involved. The authors suggest that in the presence of malonyl-CoA, CPT1C promotes the transfer of kinesin-1 from protrudin to FYCO1 on late endosomes/lysosomes, thus promoting their outward movement in neurons [228]. However, as mentioned above, this kinesin transfer model requires further testing.

Mitochondria are known to interact extensively with the ER in live cells [22], and motile mitochondria can extend ER tubules [22,38]. ER-associated mitochondria preferentially localised to acetylated microtubules [22], which are the preferred track for kinesin-1 (e.g., [229]), which is a motor for both mitochondria (e.g., [230]) and the ER. Mitochondria were also seen to interact with lysosomes, and the moving lysosome could pull out a thin tubule from the mitochondrion [38]. As similar thin tubules were found to extend from mitochondria at points of ER contact via the action of KIF5B and its mitochondrial receptor Miro1 [230], it prompts the question as to whether the PDZD8-induced three-way MCSs between ER, late endosomes, and mitochondria could be involved in both processes. However, this complex interaction essentially immobilised the organelles [126].

MCSs are clearly vitally important for many aspects of ER function, metabolism, and overall dynamics. This makes it challenging to interpret alterations seen following experiments designed to disrupt one aspect of MCS function. This is exemplified by experiments where Rab7a function—needed for late endosome/lysosome MCS, and involved in recruiting both kinesin-1 and dynein to endosomes—was disrupted. Rab7a depletion, or expression of a GDP-locked Rab7a, led to an accumulation of CLIMP63-labelled sheet-like ER at the cell periphery, and activation of the ER stress response [231]. Mateus et al. hypothesise that the structural change is caused by ER stress, rather than changes in ER dynamics [231]. Indeed, there are many ways in which stress can influence MCS and the ER (reviewed in [85]). Monitoring ER stress levels will be an important control in future studies (e.g., [219]).

#### 3.1.3. Motor-Independent ER-Microtubule Interactions

It was clear from early studies that three kinds of ER–microtubule interactions exist: motor-driven translocation of ER tubules, static interactions, and the attachment of ER tubules to growing microtubule tips [23]. The latter interaction drives tubule extension via the formation of ‘tip attachment complexes’, or TACs, first seen in *Xenopus* egg extracts [232]. In a variety of cultured cells, motor-driven sliding was seen to predominate over TACs and static interactions between ER tubule tips and microtubules [20,21,23,38], although the percentage differed between cell types [21]. TACs consist of the transmembrane ER protein STIM1 (stromal interaction molecule 1), which interacts with the MT plus-end-tracking protein EB1 [21]. STIM1 interacts with Orai1 at the plasma membrane upon depletion of ER calcium stores to permit calcium transfer from outside the cell into the ER (see Section 2.2.5). Interestingly, triggering this process prevented STIM1 tip tracking [21]. While the presence of a TAC did not change the rate of microtubule growth and shrinkage, it reduced the likelihood of the microtubule undergoing a catastrophe (starting to depolymerise) [232]. ER tubules can also be extended by attaching to a depolymerising microtubule via a dTAC [38]; the composition of ER dTACs is not known.

Why have two methods for transporting the ER towards the cell periphery? In the case of *Xenopus* embryos, although kinesin-1 is present on the ER, it is not active until later in development [187]. Instead, the ER network distributes throughout the cytoplasm via the combined activity of dynein pulling it towards the nucleus and centrosome, and TACs extending it outward as microtubules polymerise [184]. In cultured cells, TACs would ensure the ER reaches right to the cell edge. In contrast, kinesin-1’s preference for stable microtubules as tracks, perhaps because they have less MAP7 bound to their surface, means it translocates poorly to the end of newly-polymerised microtubules [229]. An important role for STIM1-EB1-mediated ER localisation has been demonstrated in neuronal growth cones, where STIM1 is needed for the orientation of microtubules and ER that is essential for a growth cone to move towards a growth factor gradient [233].

The ER can also attach statically to microtubules, either via tubule tips, junctions, or along the length of the tubule [187,232]. Static links between the ER and microtubules may be formed by the previously mentioned proteins CLIMP-63 [15], p180 [16,194], REEP1 [17], and Sec61β [234]. These links have been hypothesised to regulate ER network tension and positioning [235]. It is also possible that ER attachment may stabilise microtubules, as has been seen for p180 in axons [194]. One problem with assessing this is that over-expression of a microtubule-binding protein may cause stabilisation (e.g., [16,234]) that might not be seen at endogenous levels. However, this potential reciprocity between ER tubules and microtubules will be important to consider in the future, not least because stabilised microtubules would be ideal tracks for kinesin-1 transport of other ER tubules.

#### 3.1.4. ER Interactions with the Actin Cytoskeleton

In animal cells, microtubules play the dominant role in remodelling the ER via the surprisingly varied mechanisms described above, whereas in plants, algae, and some fungi, actin-based motility is key (reviewed in [236]). Cytoplasmic streaming, the myosin-driven directional flow of the cytoplasm, also plays a role in plant cell ER dynamics [237,238]. Interestingly, a novel ER and microtubule-associated compartment (EMAC) has been identified in plants that is anchored on microtubules, thereby resisting this flow [239]. Similarly, stable tubular ER junctions have also been observed to associate with microtubules [240]. A surprising need to anchor the ER has also been seen in *C. elegans* nematodes, where the loss of an ER and nuclear envelope-localised nesprin, ANC-1, leads to uncontrolled streaming of the ER, mitochondria, and lipid droplets due to forces generated as the worm crawls [241]. However, it is not yet clear what cytoskeletal component ANC-1 binds to.

In animal cells, myosin-V-driven ER motility may contribute to short-range ER movement (e.g., [211,242]). Furthermore, filamin proteins may bind the ER to actin in mouse embryonic fibroblasts [243] and interactions between the ER and actin filaments via myosin 1c may stabilise sheets [188]. The degree of reliance of the ER network on the actin cytoskeleton differs between cell types, since depolymerising actin filaments with cytochalasin had little effect on ER network distribution in VERO cells [20], while in the epithelial-like cell line Huh-7, actin depolymerisation resulted in smaller, less abundant ER sheets at the cell periphery and irregular ER positioning, with large regions of the cells devoid of the organelle [188]. Furthermore, when microtubules are depolymerised, a slow, actin-dependent retraction of the ER towards the cell centre is seen in some [23,186,244] but not all [20] cultured cell types, demonstrating that ER–actin interactions influence ER morphology. Importantly, super-resolution live-cell imaging has revealed microtubule-independent ER tubule extension [38], which may well be due to actin-based motility.

### 3.2. Network Fluctuations

The tubules, junctions and sheets that make up the endoplasmic reticulum fluctuate over time. Even the established elements of the network that are not being actively rearranged are not stationary, but rather oscillate around a position. Given the complex morphology and fast dynamics of the ER, quantifying ER motion is a current challenge in the field.

A widely accepted measure of dynamics is the mean squared displacement (MSD). The MSD is a measure of how far an object is expected to travel over a specified time. To calculate the MSD, firstly the difference in position between the object at an initial time and the object at a later time is found. Each pair of points separated by the same time difference (or lag time) is considered, and the mean difference in position for all of these pairs gives the MSD at that particular lag time. Mathematically, the MSD, 〈Δr^2^(τ)〉, is defined as:〈Δr^2^(τ)〉 = 〈(x(t + τ) − x(t))^2^ + (y(t + τ) − y(t))^2^〉,(1)
where x(t) and y(t) give the position of the object at time t and τ is the lag time. Fitting a power law to the MSD allows the motion to be described as diffusive, sub-diffusive, or super-diffusive. Diffusion, or Brownian motion, describes processes such as the gradual mixing of regions of high and low concentrations to form an even distribution. MSDs for diffusive motion follow 〈Δr^2^(τ)〉 ∝ τ. Anomalous diffusion is the term used to describe cases in which the MSD is not diffusive. Super-diffusion describes active transport, such as motor protein-driven motion, and the MSDs follow 〈Δr^2^(τ)〉 ∝ τ^α^ where 1 < α < 2. On the other hand, sub-diffusive motion is associated with motion in a crowded environment and results in MSDs with exponents α < 1. By tracking points in the ER network and calculating the MSDs of these trajectories, information about the dynamics of individual components of the ER has been elucidated. ER junctions have been found to move sub-diffusively, in a microtubule-dependent manner [245]. Nocodazole treatment reduced the number of diffusive and super-diffusive junctions, indicating that interactions with motor proteins may be the cause of the directed junction motion. Tubules in the ER network oscillate sub-diffusively in general, with an average MSD exponent of 0.5 [25]. This result is in agreement with the theoretical prediction of 〈Δr^2^(τ)〉 ∝ τ^1/2^ for semi-flexible polymers oscillating thermally in a viscous medium [246]. These results indicate that the movement of the network is constrained, which is consistent with motion in a crowded environment such as the cytoplasm.

Recently, the full contours of individual tubules have also been tracked, resulting in a more detailed description of ER tubule dynamics [27]. The distributions of perpendicular displacements from the mean tubule backbone position were found for points along the tubule contour. The asymmetry, or skewness, of these distributions was calculated. Significant skewness indicated active, driven dynamics, whereas positions with negligible skewness described points oscillating thermally, without any active influences. The majority of points oscillated thermally, however, one-third of the points tracked fluctuated actively. Dynamic constrictions and bulges in the diameter of ER tubules have also been observed [38,181], although the characteristics of these oscillations are yet to be analysed in depth. As a result of both the MSD and contour tracking studies, ER tubules are thought to oscillate sub-diffusively in general, with active dynamics occurring occasionally along the tubule backbone. The sub-diffusive fluctuations are likely to be caused by the motion and crowding of the cytoplasm, whereas active dynamics may be caused by motor proteins or membrane contact sites with trafficking organelles.

Sheet dynamics have not been studied in as much detail as tubular dynamics. This is probably due to the difficulty in quantifying the motion of lamellar objects, particularly those within living cells. In particular, lamellar regions of the ER are often located in the thicker parts of the cell near the nucleus, in which out-of-focus light from other imaging planes makes capturing a video showing just one layer of ER sheets challenging. Even with clear videos showing ER sheets, quantifying the dynamics of lamellar membranes is challenging due to the lack of accepted methods to describe sheet dynamics. Two possible measures of sheet dynamics are the velocity of sheet edges and the out-of-plane fluctuations of sheets. To our knowledge, neither of these approaches have yet been implemented for perinuclear ER sheets in vivo. Theoretical predictions have been made for the out-of-plane fluctuations of membrane sheets. The MSD of these undulations is predicted to increase as τ^2/3^ for free membranes and more slowly, as ln(τ), for membranes under tension [246].

Experimentally, some studies have considered the dynamics of lamellar ER at the cell periphery. Joensuu et al. [188] discovered that the location and dynamics of peripheral ER sheets are dependent on actin filament arrays and foci. ER sheets were found to dynamically rearrange in response to the movement of actin structures. The disappearance of actin led to ER sheets filling in the space left behind and the formation of new actin structures caused the opening of a fenestration in an ER sheet. The dynamics of sheet edges were also studied. Sheets fluctuated in a small area and showed no preference for direction in untreated cells. Treatment with the actin polymerisation inhibitor, latrunculin A, increased the lateral movement of sheets as well as the proportion of sheets undergoing fission, fusion, or transformations into tubules.

The studies detailed in this section show that the fluctuations of established structures within the ER are complex, varied and influenced by many subcellular organelles and processes. The interplay between ER dynamics and the dynamics of other subcellular organelles and structures is only just beginning to be understood and fruitful research in this area is expected in the near future.

### 3.3. Dynamics of Membrane and Lumenal Components

The processes carried out by the ER involve an abundance of transmembrane and lumenal proteins, many of which move in order to seek out interacting partners. The dynamics of these proteins, as well as the lipids forming the ER membrane may affect the overall dynamics of the organelle.

Fluorescence techniques such as single-particle tracking (SPT), fluorescence recovery after photobleaching (FRAP), and fluorescence correlation spectroscopy (FCS, reviewed in [247]) have been used to quantify lipid and protein motion. Computer simulations have also been used to study the dynamics of objects embedded in membranes, as the fluorescent probes used to track subcellular objects are thought to hinder dynamics. High concentrations of the fluorescent dye Rhodamine are proposed to cause hydrodynamic drag, decreasing the diffusion coefficient of the objects of interest by up to 20% [248].

Several membrane properties are known to influence the dynamics of transmembrane and lumenal components: lipid rafts, protein concentration, protein folding status, cytoskeletal interactions, and membrane tension. Lipid rafts are domains of clustered lipids and proteins that move within the bilayer [249]. Diffusion was found to be slower by a factor of two within lipid rafts [250], and lipids and proteins can become transiently confined to these rafts, in which a hindered, sub-diffusive motion was observed [251]. Higher concentrations of proteins within the lipid bilayer are also known to slow lateral diffusion [252], with simulations concluding that lateral diffusion in highly crowded membranes was a factor of 5-10 slower than in dilute membranes [253]. FCS experiments also showed that the folding status of transmembrane proteins affects their motion within the ER membrane. Several proteins were analysed, all of which were found to move sub-diffusively [254]. The anomalous exponent of unfolded VSVG was found to be lower than that of its folded form. This highlights the more obstructed dynamics of unfolded proteins. The binding of calnexin, a transmembrane chaperone protein, to unfolded VSVG caused an increase in the anomalous exponent such that the motion was indistinguishable from the folded form. This result indicates that calnexin may prevent the formation of harmful immobile structures of unfolded proteins.

Collisions between the cytoplasmic domains of transmembrane proteins and cytoskeletal filaments are also known to slow lateral movement within lipid bilayers [255], as has been shown for transferrin receptor (TfR) at the plasma membrane. Under normal conditions, slow, confined motion of TfR was observed; when actin was depolymerised with latrunculin, free diffusion was observed [256]. Photoactivation experiments in tobacco leaf epidermal cells however found that transmembrane proteins in the ER exhibited slower, diffusive dynamics when treated with latrunculin B in comparison to the active dynamics observed in untreated cells [257]. This is most likely due to the myosin-driven reorganisation of the ER in plant cells (Section 3.1.4). Another example of transmembrane protein dynamics being altered by cytoskeletal interactions is the motion of ER exit sites. ERES move sub-diffusively along ER tubules in a microtubule-dependent manner [61,180]. Lower anomalous exponents and smaller diffusion coefficients were measured when cells were treated with nocodazole, indicating that microtubular activity promotes ERES dynamics. In simulations, applying tension to the membrane, as would happen with motor activity, increased the lateral diffusion coefficients of lipids in the bilayer, without altering their anomalous exponents [258]. The anomalous exponents were sub-diffusive, with a value of ~0.75 observed for all membrane tensions. The dynamics were also found to be dependent on the direction in computer simulations. Deviations in the direction perpendicular to the bilayer were found to be constrained, whereas lateral motion in the plane of the bilayer was not [259].

Taken together, these results show that the dynamics of membrane lipids and transmembrane proteins are complex and depend on the composition and state of the lipid bilayer, and upon interactions with the cytoskeleton.

The dynamics of substrates within the lumen of the ER have also been measured experimentally. Translational diffusion of proteins within the lumen of the ER was first experimentally explored using green fluorescent protein (GFP) in 1999 [260]. The motion of GFP in the ER lumen was found to be significantly slower than in the cytoplasm and in mitochondria. The dynamics of calreticulin, a lumenal chaperone protein, were found to depend on the folding environment of the ER [261]. In quiescent cells, calreticulin was found to readily sample the whole ER, whereas slower diffusion coefficients were observed in actively metabolising cells. Single-particle tracking experiments revealed that both calreticulin and ER-targeted lumenal HaloTag proteins moved with slower velocities at ER junctions than in tubules [181]. The faster population was diminished upon ATP depletion, indicating that the ATP-dependent motor protein-mediated dynamics of the ER may contribute to the dynamics of lumenal components. This velocity difference between tubules and junctions was not observed for the transmembrane chaperone calnexin. Treatment of Cos-7 cells with latrunculin B led to faster lumenal protein dynamics, as did removing N-glycans from the proteins of interest [262]. This study, along with the experiments using TfR described above [256], indicate that actin may play a major role in governing the motion of proteins and lipids in the lumen and membrane of the ER.

A causal connection between the motion of the ER and the motion of lumenal or membrane-bound components is yet to be made. However, several hypotheses have been proposed. Georgiades et al. proposed that the oscillations of ER tubules promote the mixing of reactants within the lumen, speeding up the biosynthetic processes performed in the ER [25]. Similarly, Stadler et al. suggest that the motion of ERES is caused by ER network fluctuations, which are strongly microtubule-dependent [180]. Holcman et al. take a different approach and propose that ATP-dependent peristalsis-like contractions in ER tubule diameters generate active transport of lumenal components [181]. It is clear that the movement of the ER and the dynamics of ER-resident components are linked, and it is thought that the motion of the ER increases the efficiency of cellular processes carried out in the organelle, although this has not yet been confirmed.

### 3.4. Computational Analysis of ER Dynamics

Analysing the dynamics of the ER in vivo is challenging due to its complex morphology and rapid dynamics as well as the phototoxicity of fluorescence microscopy experiments. Mathematical modelling of the network is an opportunity to gain further insights into this complex system from a different viewpoint. ER tubules can be regarded as semi-flexible polymers, with junctions represented by cross-links. Recently, many groups have modelled the dynamics of such cross-linked networks of semi-flexible polymers [263,264,265,266], including those influenced by active forces such as molecular motors [267,268,269,270]. Such models may provide insights into ER network dynamics by simplifying the system and only considering interactions with microtubules. A recent study modelling the motion of a particle surrounded by a network of semi-flexible polymers may provide insights into the functional reason for active rearrangements of the ER. Gong et al. [270] found that diffusion of a particle embedded in a network of semi-flexible polymers was enhanced by the presence of active motors. Diffusion of particles in a network with active motors was super-diffusive, whereas, without active motors, sub-diffusion was observed. A possible explanation given for this enhanced motion was the greater number of collisions between the particle and the network in the presence of active motors. This result may relate to the dynamics of particles in the lumen of the ER. Fluctuations in the tubules and sheets of the network may induce faster dynamics of the particles within the lumen due to more frequent collisions with the lumenal face of the lipid bilayer.

It has also been shown that certain aspects of plant ER morphology and dynamics can be replicated by considering the ER as a minimal network [271,272,273], where points are connected using the minimum length of tubules possible. This model accurately recreates aspects of ER morphology and dynamics. The structure of tubular regions of the network is accurately recreated, as are the dynamics of ring closure. This approach, however, is as yet unable to relate the rapid rearrangement dynamics of the network found using the model to those observed experimentally [273].

The motion of lumenal particles has also recently been simulated. A model describing the dynamics of particles within the lumen of the ER searching for reaction partners concluded that the morphological properties of the network are the most influential factors in determining the exploration times of molecules [274]. Future improvements in imaging data, as well as in the models used to represent the ER network, will result in improvements in both the conclusions of computational models and in workflows for analysing experimental data.

Recently, several groups have released open-source software designed to extract dynamic variables from videos of the ER in live cells [24,27]. Such methods to quantify the morphology and dynamics of the ER network are essential to advance our understanding of this organelle. Analysing the changes in dynamics and morphology under various cellular conditions will shed light on the underlying reason for ER dynamics and how altered dynamics may lead to clinical pathologies.

## 4. Morphology, Dynamics & Disease

It is unsurprising that an organelle such as the ER, with its critical and diverse functions, would be involved in many diseases. Dysregulation of the morphology and distribution of the functional subdomains of the ER (as described in Section 2.2) is therefore likely to affect their function and may result in disease states, particularly in neurons [32]. In fact, many ER morphology-regulating proteins are known to be mutated in human diseases [227]. Miscommunication between the ER and mitochondria has also been identified as a contributor to many diseases, either via dysregulation of calcium ion transfer, insulin signalling, or mitochondrial division. Interactions between the ER and endosomes/lysosomes are similarly vital for normal cellular function, with particular problems being seen in neurons when proteins involved in these interactions are mutated [195,227]. Additionally, many organisms that give rise to pathogenic infections rely on ER machinery for their replication or entry into the cell. The best-characterised proteins involved in both diseases and in ER maintenance are summarised in Table 1. Another major factor in many diseases, particularly cardiovascular diseases, is ER stress. MCSs are affected by many other types of stress [85,275], and so this is likely to prove an important factor in disease. Further work is needed to fully explore the interplay between ER morphology and ER stress. Links between excessive ER stress, UPR activation, and the onset of diseases were recently reviewed [276,277,278], but are not addressed here.

ER morphology-regulating proteins are implicated in many diseases, including multiple neurodegenerative disorders (see [227] for an excellent, thorough review). Atlastins, a family of proteins that mediate the fusion of ER tubules to the network [4,348], are known to be mutated in hereditary spastic paraplegia (HSP) [285] and hereditary sensory and autonomic neuropathy (HSAN) [346,347]. REEPs and protrudin, which are known to stabilise regions of high membrane curvature in the ER [2,281], are also mutated in some forms of HSP [280,286], suggesting that abnormal ER morphology plays a role in the disease.

ER-late endosome/lyosome MCSs are frequently seen in neurons [70] and are clearly vitally important given that they are commonly affected by disease-causing mutations (Table 1; [227]). Protrudin mutations may affect ER distribution in axons, due to its interactions with kinesin-1, and its role in generating motile ER tubule-late endosome MCSs [205]. This function is vital for efficient neurite extension [129,195,204,227]. Mutations in KIF5A could conceivably affect this pathway. The disruption of ER–endosomal interactions also affects endosomal sorting and leads to lysosomal defects (Section 2.2.6 and 3.1.2), and this is frequently linked to diseases such as HSP and ALS (Table 1), where it can be caused by mutations in spastin, strumpelin, REEP1 [177], and VAPB, as described elsewhere in this special issue [291].

Mutations in proteins regulating ER morphology and dynamics have also been associated with amyotrophic lateral sclerosis (ALS), Alzheimer’s disease, Warburg Micro syndrome and spinocerebellar ataxia type 2. Reticulon 3 and 4 are mutated in Alzheimer’s disease [336] and ALS [299], respectively. Reticulons regulate ER membrane curvature [2] and their mutation in ALS changes the distribution of chaperone proteins within the ER [299] and therefore is likely to negatively affect ER function. Recently, Mookherjee et al. discovered that protein aggregation in the cytoplasm, a common hallmark of neurodegenerative diseases, affects both ER morphology and dynamics [359]. Reticulon 4 was found to bind to the cytoplasmic aggregates, which may be the cause of the abnormal ER structure observed. Fewer three-way junctions, slower lumenal dynamics, and impaired tubule fusion efficiency were observed in cells with protein aggregates. Therefore not only do morphology-regulating proteins play a role in the onset of the disease, but they may also influence cellular processes once the disease has progressed.

Warburg Micro syndrome, a rare disorder that results in neurodevelopmental defects, can be caused by mutations in Rab18 [29]. Rab18 is involved in several ER-related processes including the regulation of ER-lipid droplet MCS involving the NRZ complex and SNARE proteins [343,360] (but not under all conditions [8]) and the maintenance of normal ER morphology [8]. As described in Section 3.1.1, Rab18 has also recently been shown to promote anterograde ER tubule transport via its interaction with kinectin-1 [198] and in agreement with this result, the dynamic tubular regions of the network were absent in cells depleted of Rab18 [8]. A similar relationship between morphology-regulating proteins, ER dynamics and disease is observed with ataxin-2 in spinocerebellar ataxia type 2 (SCA2). Mutations in ataxin-2 are known to cause SCA2 [350]. Depletion of ataxin-2 increased the proportion of sheets in the ER network as well as hindering both tubular and lumenal dynamics [28]. Unidirectional, long-distance translocation of tubules was diminished and FRAP experiments showed that the transport of lumenal components was much slower in the absence of ataxin-2 [28]. Mutations in Rab18 and ataxin-2, proteins known to play a role in extending ER tubules, both cause defects in tubular morphology as well as dynamics.

Alongside membrane-shaping proteins and proteins involved in ER-late endosome/lysosome MCSs, another likely cause for diseases is the miscommunication between the ER and mitochondria. ER–mitochondria MCS are important sites of Ca^2+^ ion transfer [275]. Impaired calcium homeostasis was first proposed as a possible cause for neurodegenerative diseases by Khachaturian in 1994 [361]. Since then, mutations in many proteins regulating ER–mitochondria MCS have been implicated in human diseases. Mutations have been found in six proteins that are either recruited to, or involved in forming ER–mitochondrial MCSs in Parkinson’s disease: Miro1 [318], Parkin [321], DJ-1 [323], α-synuclein [325], PINK1 [327], and LRRK2 [329]. These mutations were all found to alter calcium ion transfer between the organelles, disrupt ER–mitochondrial MCS, or both (see Table 1). It is important to note that Miro1 also recruits kinesin-1 to mitochondria, and KIF5A mutations have been shown to affect mitochondrial motility [362]. In Alzheimer’s disease, mutations in presenilins also perturb calcium ion homeostasis [335]. In this case, however, it is the operation of ryanodine receptors (RyR) and inositol 1,4,5-triphosphate receptors (IP_3_R) that are dysregulated, rather than the MCSs [333,334,363]. Charcot Marie Tooth (CMT) disease has also been linked to mutations in ER–mitochondria MCS-forming proteins [303,306], although no link has been found between CMT and calcium dynamics.

The dysregulation of membrane contact sites between the ER and mitochondria has also been implicated in other non-neurodegenerative diseases. Impaired insulin signalling, a hallmark of diabetes, is observed in cells with mutations in mitofusin 2 [308,309] and the VDAC-1/grp75/IP_3_R-1 Ca^2+^ transfer complex [310], both of which tether the ER and mitochondria at MCSs. Additionally, many proteins at the ER–mitochondria interface have been implicated in various cancers (reviewed in [345]). Abnormal mitochondrial fission at ER–mitochondrial MCSs due to mutations in dynamin-related protein 1 also lead to encephalopathy [340]. Additionally, ER–mitochondrial MCSs have been linked to ALS via mutations in the sigma-1 receptor [300], which is involved in Ca^2+^ homeostasis at MCSs. These MCSs were disrupted in mutant cells [302]. ALS mutations have also been linked to altered Ca^2+^ dynamics at ER–mitochondrial MCSs. The ALS mutant form of VAPB leads to an altered MCS with the outer mitochondrial membrane protein PTPIP51, perturbing Ca^2+^ uptake at these MCSs [296]. Many other ALS mutations affect ER–mitochondrial contacts, such as those in VAPB (see reviews by Chen et al. and Borgese et al. in this issue [275,291]).

Intracellular calcium homeostasis also involves MCSs between the ER and the plasma membrane. Upon depletion of ER calcium stores, extracellular Ca^2+^ is transferred across the plasma membrane into the ER at MCS via SOCE. The ER-resident protein STIM1 and the plasma membrane-resident protein Orai1 form a complex, facilitating SOCE (see Section 2.2.4). Mutations in either, or both of these proteins are known to cause tubular aggregate myopathy (TAM) and the related condition Stormorken syndrome via dysregulation of calcium homeostasis [311,312,313,314,315]. Since STIM1 binding to EB proteins generates TACs, it will be interesting to determine the effects these mutations have on TAC-based ER tubule extension.

Pathogenic infections are also linked to ER-related proteins. Pathogens hijack the ER in the host cell to further their replication (reviewed in [364,365]). For several of these pathogens, proteins have been discovered that bind to ER-resident proteins in order to form MCSs. These sites are crucial for pathogen replication. Known ER-interacting pathogens include *Legionella pneumophila* [352], Brome mosaic virus (BMV) [356], the non-enveloped polyomavirus SV40 [206], enterovirus 71 [357], flaviviruses such as the Zika virus [358], and *Chlamydia trachomatis*. Reticulons are targeted by *Legionella pneumophila* via the binding of Ceg9 [352], BMV via viral protein 1a [356], and enterovirus 71 via enterovirus protein 2C [357]. Pathogen replication is promoted by the binding of reticulons in all three of these cases [351,356,357]. *Legionella pneumophila* also interacts with atlastins [352] to promote replication [351], as do the flavivirus family members: Dengue virus, Zika virus, and West Nile virus [358]. *Chlamydia trachomatis* proteins bind to VAPA, VAPB [353], and CERT [354] to form membrane contact sites with the ER. Depletion of CERT or the VAP proteins diminished bacterial replication [354], demonstrating once again that ER-pathogen MCSs are critical for pathogen replication. Interestingly, ER membrane shaping proteins have also been found to suppress viral replication. Reticulon 3 was found to bind to non-structural protein 4B (NS4B) of the hepatitis C virus [366]. The self-oligomerisation of NS4B facilitates viral replication [367]. However, when reticulon 3 is bound, self-oligomerisation of NS4B is prevented and therefore viral replication is suppressed [366]. SV40 enters the cytoplasm by penetrating through the ER membrane using the ERAD machinery [368], most likely in an expanded perinuclear ERQC (Section 2.2.1), and this has been suggested to involve kinesin-1 function via binding to B14, an ER-localised DNA-J domain-containing membrane protein [206].

In summary, mutations in single proteins can have catastrophic effects on ER morphology, dynamics, and MCSs, leading to clinical pathologies. In many cases, morphology, dynamics, and MCSs are tightly linked, with mutations affecting one of these three attributes often causing knock-on effects for the others. Recent research into disease-causing mutations is beginning to consider ER morphology and dynamics as related topics, as opposed to viewing them independently. Future work in this area is likely to provide insights into the relationship between morphology and dynamics as well as how ER dynamics are involved in diseases that are known to alter morphology.

## 5. Discussion

Many proteins that are responsible for regulating ER morphology have been identified and the links between morphology and function are now becoming clear. We are only just beginning to quantify ER dynamics and to understand what effect the dynamics may have on the processes performed by the organelle. Mutations in morphology-regulating proteins have been found to cause many human diseases (Table 1) and in many cases, the structure of the ER was abnormal, indicating that the morphology, dynamics, and function of the ER are intrinsically linked. ER dynamics are also clearly vitally important in establishing and maintaining the organisation of the ER and its many MCSs. Again, the proteins involved are commonly mutated in disease. A key challenge for the future is to define the roles played by these dynamics, and tease apart the contribution of different kinds of motility. This will need faster frame rate imaging that has commonly been used, since highly motile ER tubules and potentially interacting organelles move at speeds of up to 5 µm/s, and cannot be reliably detected when imaging at one frame every 1.5–5 s, as have commonly been used (e.g., [22,38,75,176,219]). It is also important to note that the go-to cell line for studying ER dynamics, the Cos-7 cell, has a much less dynamic ER than other cell lines [20]. The advent of super-resolution microscopy of fixed and live cells will continue to provide a wealth of new understanding of ER structure and dynamics. We expect that with the recent publication of several methods to analyse and quantify ER dynamics, more links between dysregulated ER dynamics and disease will be discovered. Hopefully, these discoveries will result in further understanding of the functional basis for ER movement. The future of ER dynamics research looks bright!

## Figures and Tables

**Figure 1 cells-10-02341-f001:**
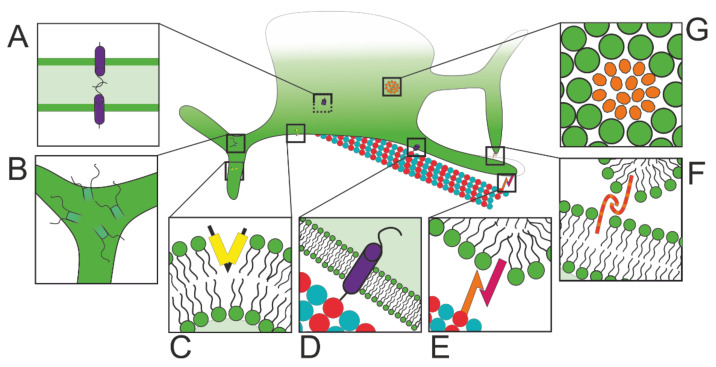
Diagram showing the possible mechanisms of action of ER morphology-regulating proteins. (**A**) CLIMP-63, a transmembrane ER protein, may regulate sheet thickness by forming a dimer, linking the two bilayers of an ER sheet. (**B**) Lunapark locates to and stabilises three-way junctions in the ER network, although how this is achieved is not yet known. (**C**) Proteins promoting membrane curvature, such as the reticulon and REEP families, are shaped like hairpins and localise to tubules and sheet edges (regions of high membrane curvature). The proteins are embedded in the outer leaflet of the bilayer, with the wider end at the surface, inducing membrane curvature. (**D**) Static interactions between the ER and microtubules may be facilitated by CLIMP-63, p180, REEP1, and Sec61β. These proteins are all resident in the ER membrane and possess microtubule-binding domains. (**E**) Tip attachment complexes (TACs) are formed by interactions between the ER-resident protein STIM1 (pink) and the microtubule plus-end tracking protein EB1 (orange). (**F**) New tubules are fused to the network by atlastin, an ER-resident GTPase, which may reside in both of the sections of ER membrane to be fused. A dimer may be formed between the two proteins, promoting tubule fusion. Rab10, Rab18, and Drp1 may also promote tubule fusion, although the mechanisms are currently unknown. (**G**) Polyribosomes, groups of ribosomes gathered on the surfaces of ER sheet membranes, may maintain sheet flatness, although this has yet to be experimentally demonstrated.

**Figure 2 cells-10-02341-f002:**
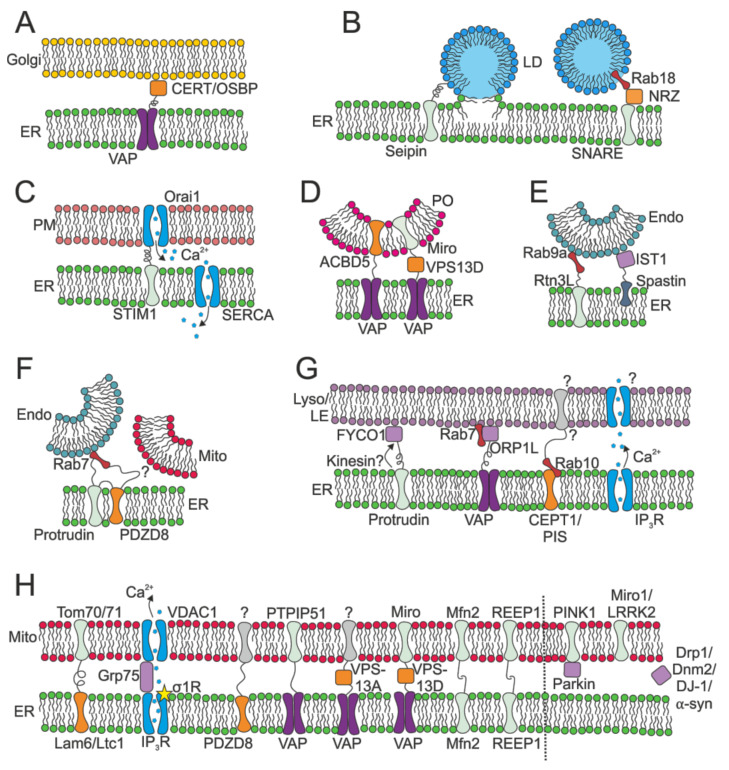
Membrane contact site proteins that are discussed in this review. MCSs between the ER and the Golgi apparatus (**A**), lipid droplets (LD, (**B**)), the plasma membrane (PM, (**C**)), peroxisomes (PO, (**D**)), early endosomes (Endo, (**E**)), late endosomes/lysosomes (LE/Lyso, (**G**)) and mitochondria (Mito, (**H**)) are shown. The triple MCS between the ER, endosomes, and mitochondria is also depicted (**F**). Orange proteins are those known, or thought to be, involved in lipid transfer and blue proteins are calcium ion channels. Proteins with no clear function in relation to lipid or calcium ion transfer are coloured light green (transmembrane) or lilac. The dotted line in H separates MCS-forming proteins (on the left of the line) from proteins recruited to MCSs and those whose interacting partners are unknown or not included in this review. The ER–mitochondrial MCS-forming mechanisms of mitofusin 2 and REEP1 are still unclear, however, both proteins may form homodimers in order to tether the organelles [105,106].

**Figure 3 cells-10-02341-f003:**
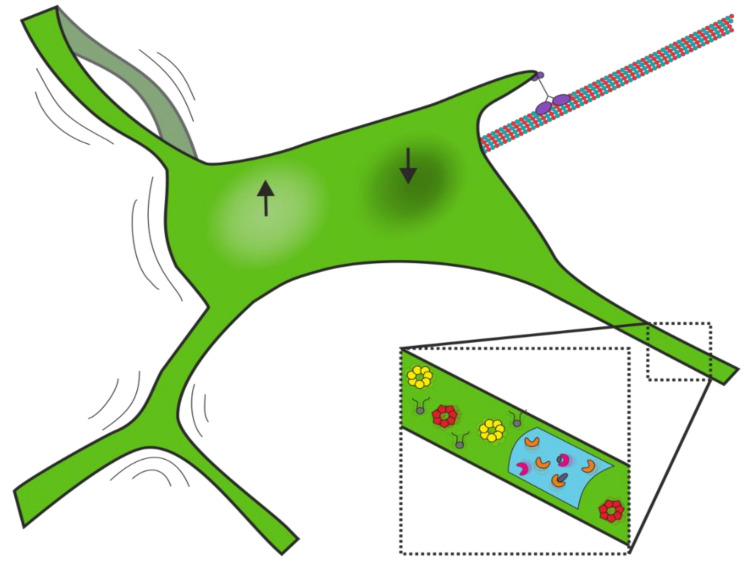
Schematic depicting the dynamics of the ER network. Fluctuations of the tubules, junctions, and sheets of the ER are shown with black lines and arrows. Tubules, junctions, and sheet edges oscillate laterally (within the plane of the page) and vertically (perpendicularly to the plane of the page). Vertical sheet fluctuations, as shown by the black arrows, are also thought to occur. The transmembrane and lumenal proteins also move, as shown in the inset. Motor proteins bind to the ER and move along microtubules to draw out new tubules from the existing network (see top right).

**Figure 4 cells-10-02341-f004:**
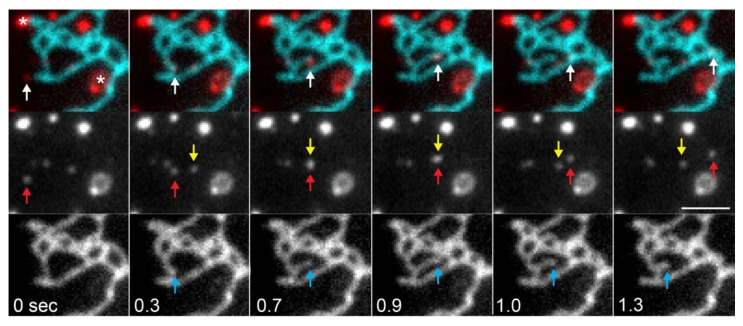
Early endosomes associate dynamically with the ER. A small endosome (red and white arrows) transiently associates with the ER and extends an ER tubule (blue arrow). When released from the endosome, the ER tubule retracts. The moving endosome/ER tubule briefly interacts with another endosome (yellow arrow). Other endosomes interact statically with ER tubules (asterisks). Images of GFP Rab5 (pseudocoloured red for easier visualisation of the small vesicles) and mCherry ER marker (LongER, [20]) (cyan) were collected simultaneously at 10 fps in wide-field mode on a DeltaVision OMX (A*STAR Institute of Medical Biology, Singapore) by V. Allan. Scale bar = 2 microns.

**Table 1 cells-10-02341-t001:** Diseases related to ER morphology and dynamics. The proteins mutated in the disease or related to the replication of the pathogen are listed, alongside their normal functions in the ER and how their function is affected/hijacked in the disease.

Disease	Protein Implicated	Protein Role in Healthy ER	Function Affected in Disease
Hereditary Spastic Paraplegias	Spastin [279]	ER tubule regulation [17], ER-endosome MCS [177]	Endosomal fission fails at MCS [177]
	Protrudin [280]	ER membrane curvature and tubule fission [281], ER-endosome MCS [204], ER MCS-dependent endosome maturation [125]	ER morphology and function dysregulated [225,281]
	Seipin [282]	ER-lipid droplet MCS [283]	Axon regeneration impaired [284]
	Atlastin-1 [285]	ER membrane fusion [5]	ER morphology dysregulated [17]
	REEP1 [286]	ER membrane curvature [2], ER-mitochondria MCS [106]	ER-mitochondria interactions disrupted [106]
	REEP2 [287]	ER membrane curvature regulation [2]	ER morphology disrupted [288]
	KIF5A [195]	ER dynamics via kinesin-1 [195]	Impaired axonal transport [195]
Amyotrophic Lateral Sclerosis	VAPB [289]	MCS between ER and many organelles (reviewed [290,291])	ER morphology dysregulated [292,293,294,295], ER-mitochondria MCS [296,297],interactions between VAPB and oxysterol binding protein (OSBP) perturbed [298]
	Reticulon 4 [299]	ER membrane curvature [2]	Chaperone protein disulfide isomerase (PDI) distribution altered [299]
	Sigma-1 Receptor [300]	ER-mitochondria MCS [301], ER Ca^2+^ homeostasis [301]	Disrupts ER-mitochondria MCS [302]
	KIF5A [195]	ER dynamics via kinesin-1 [195]	Impaired axonal transport [195]
Charcot Marie Tooth	Dynamin 2 [303]	Possibly involved in mitochondrial fission at ER-mitochondria MCS [304,305]	Unknown
	Mitofusin 2 [306]	ER-mitochondria MCS [105]	Reduction in ER-mitochondria MCS [307]
	KIF5A [195]	ER dynamics via kinesin-1 [195]	Impaired axonal transport [195]
Diabetes	Mitofusin 2 [308]	ER-mitochondria MCS [105]	Insulin signaling impaired [309]
	VDAC-1/grp75/IP_3_R-1 [310]	ER-mitochondria MCS [165], ER Ca^2+^ homeostasis [165]	Diminished ER-mitochondria interaction impairs insulin signaling [310]
Tubular Aggregate Myopathy/Stormorken Syndrome	STIM1 [311,312,313]	SOCE [168], ER-plasma membrane MCS [170], TACs [21]	Ca^2+^ homeostasis dysregulated [311,313,314]
	Orai1 [315]	SOCE and ER-plasma membrane MCS [316,317]	Ca^2+^ homeostasis dysregulated [315]
Parkinson’s Disease	Miro1 [318]	ER-mitochondria MCS [319]	Altered ER-mitochondria Ca^2+^ transfer [320]
	Parkin [321]	ER-mitochondria MCS [322]	Altered ER-mitochondria Ca^2+^ transfer [322]
	DJ-1 [323]	ER-mitochondria MCS [324]	Disrupts ER-mitochondria MCS and Ca^2+^ transfer [324]
	α-synuclein [325]	ER-mitochondria MCS [326]	Fewer ER-mitochondria MCS [326]
	PINK1 [327]	ER-mitochondria MCS [328]	Altered ER-mitochondria Ca^2+^ transfer [328]
	LRRK2 [329]	ER-mitochondria MCS [330]	Disrupts ER-mitochondria MCS [330]
Alzheimer’s Disease	Presenilins [331]	ER Ca^2+^ homeostasis [332]	Ca^2+^ homeostasis dysregulated [333,334,335]
	Reticulon 3 [336]	ER morphology regulation, Golgi to ER trafficking [337]	Amyloid plaque formation altered [336,338], cognitive dysfunction due to Rtn3 aggregates in AD brains [339]
Encephalopathy	Dynamin-related protein 1 [340]	ER-mitochondria MCS [73]	Defective mitochondrial fission [340]
Retinal Dystrophy	ACBD5 [341]	ER-peroxisome MCS [136,342]	Unknown
Warburg Micro Syndrome	Rab18 [29]	ER-lipid droplet MCS [343], ER morphology and dynamics [8,198]	Perturbed autophagy [344]
Cancer	Various ER-resident and ER-mitochondria MCS proteins (reviewed [345])	ER-mitochondria MCS and Ca^2+^ dynamics implicated	Various
Hereditary Sensory and Autonomic Neuropathy	Atlastin-3 [346,347]	ER tubule fusion [348]	Aberrant bundling of ER tubules [349]
Spinocerebellar Ataxia Type 2	Ataxin-2 [350]	ER morphology and dynamics [28]	ER morphology collapse, ER dynamics impaired [28]
Legionnaires’ Disease	Atlastin-3 [351]	ER tubule fusion [348]	Atlastin-3-mediated ER remodeling promotes bacterial replication [351]
	Reticulon 4 [352]	ER membrane curvature [2]	Connects ER to pathogen by binding Ceg9 [352] to promote bacterial replication [351]
Chlamydia	VAPA & VAPB [353]	MCS between ER and many organelles (reviewed [84])	Chlamydia protein IncV binds VAP to form ER-bacterial inclusion MCS [353]
	CERT [354]	ER-Golgi ceramide transfer [117]	Chlamydia protein IncD binds to CERT to form ER-bacterial inclusion MCS [354]
	STIM1 [355]	SOCE [168], ER-plasma membrane MCS [170], TACs [21]	STIM1 localises to ER-bacterial inclusion MCS [355]
Brome mosaic virus	Reticulons [356]	ER membrane curvature [2]	Reticulons bind viral protein 1a to promote viral replication [356]
Enterovirus 71	Reticulon 3 [357]	ER morphology regulation, Golgi to ER trafficking [337]	Enterovirus protein 2C binds Rtn3 to promote viral replication [357]
Flaviviruses (Dengue, Zika, West Nile)	Atlastin-2 and -3 [358]	ER tubule fusion [348]	Atlastins promote viral replication via distinct mechanisms [358]

## Data Availability

Not applicable.

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
