# Peer review of "Intertwined and Finely Balanced: Endoplasmic Reticulum Morphology, Dynamics, Function, and Diseases"

_cells, 2021, doi:10.3390/cells10092341_

Round 1

Reviewer 1 Report

This is an interesting, thorough and very well-written review focusing on a not very frequented topic, which is ER dynamics. I believe it could be improved with the following suggestions:

1) Section 2.1.  A figure is needed to illustrate the roles and localization of the proteins mentioned in the ER subdomains.

2) In Section 2, especially when describing the protein factory (2.2.1), the classical term Rough ER should be mentioned, to connect with prior literature.

3) Section 2.2 describes ER functional subdomains in several subsections. There is no mention of the functions of ER quality control, ERAD and unfolded protein response. Subdomains such as the ER quality control compartment, ER whorls and Lewy bodies should be added in an additional subsection. Some of these structures should also be mentioned in section 4 in the context of disease.

4) Another figure should be added relating to sections 2.2.3 to 2.2.6, illustrating the different MCS and the proteins involved.

5) Fig. 1 is not very meaningful. It should be removed or added as a panel in Fig. 2.

Author Response

We are pleased to submit our revised manuscript to the Cells special issue, ‘Dysmorphia and Dysregulation of the Endoplasmic Reticulum in Degenerative Diseases’. We would like to thank the reviewers for their comments, which we have dealt with as outlined below, definitely improving the manuscript.

Reviewer 1.

This is an interesting, thorough and very well-written review focusing on a not very frequented topic, which is ER dynamics. I believe it could be improved with the following suggestions:

1) Section 2.1.  A figure is needed to illustrate the roles and localization of the proteins mentioned in the ER subdomains.

This is added as figure 1, which is referred to in the text.

2) In Section 2, especially when describing the protein factory (2.2.1), the classical term Rough ER should be mentioned, to connect with prior literature.

We have used the term rough ER now in several places. Thank you for pointing that out.

3) Section 2.2 describes ER functional subdomains in several subsections. There is no mention of the functions of ER quality control, ERAD and unfolded protein response. Subdomains such as the ER quality control compartment, ER whorls and Lewy bodies should be added in an additional subsection. Some of these structures should also be mentioned in section 4 in the context of disease.

Thank you for the suggestion. Although the fundamental details of these processes are outside the scope of our article, we had not appreciated that spatial aspects of these pathways before. We have therefore expanded section 2.2.1 to include quality control, ERAD and the ERQC. We felt these went together well with protein synthesis. This section also mentions ER whorls, but we did not find any obvious link to Lewy bodies. We have referred to this expanded section in the introduction to section 4, and mentioned the ERQC in the context of SV40 entry. We have also added citations to several other excellent reviews in the special issue to table 1, and included them in the text of section 4.

4) Another figure should be added relating to sections 2.2.3 to 2.2.6, illustrating the different MCS and the proteins involved.

This is now figure 2, with some additional references added to section 4.

5) Fig. 1 is not very meaningful. It should be removed or added as a panel in Fig. 2

We have kept this figure as figure 3, as we think it is useful for understanding the detailed ER dynamics that we discuss in section 3.2, especially for biophysicists.

Reviewer 2 Report

The review by Perkins and Allan is a very well written and comprehensive compendium on ER. I really appreciated to read it since it perfectly describes the dynamicity of the ER both in terms of morphology and function.

I only suggest including a couple of additional figures to make it more appealing from the graphical point of view.

Author Response

Thank you for the kind comments. Two new figures have been added.